# Study protocol for a mixed methods approach to optimize colorectal cancer screening in Malaysia: Integrating stakeholders insights and knowledge-to-action framework

**Diane Woei-Quan Chong** [1]*, **Vivek Jason Jayaraj**[2], **Fathullah Iqbal Ab Rahim**[3], **Sharifah Saffinas Syed Soffian**[4], **Muhammad Fikri Azmi**[5], **Mohd Yusaini Mohd Yusri**[6], **Ahmad Shanwani Mohamed Sidek**[7], **Norfarizan Azmi**[8], **Rosaida Md Said**[9], **Muhammad Firdaus Md Salleh**[10], **Norasiah Abu Bakar**[11], **Hamiza Shahar**[12], **Rima Marhayu Abdul Rashid**[13], **Shazimah Abdul Samad**[13], **Zanita Ahmad**[13], **Mohd Safiee Ismail**[13], **Adilah A. Bakar**[14], **Nor Mashitah Hj Jobli**[14], **Sondi Sararaks**[15]

1 Institute for Health Systems Research, National Institutes of Health, Centre for Health Services Research, Ministry of Health Malaysia, Shah Alam, Malaysia, 2 National Institutes of Health, Sector for Biostatistics and Data Repository, NIH Manager's Office, Ministry of Health Malaysia, Shah Alam, Malaysia, 3 Institute for Health Systems Research, National Institutes of Health, Centre for Health Equity Research, Ministry of Health Malaysia, Shah Alam, Malaysia, 4 Kedah State Health Department, Kota Setar Health District Office, Ministry of Health Malaysia, Alor Setar, Malaysia, 5 Disease Control Division, Ministry of Health Malaysia, Putrajaya, Malaysia, 6 Bandar Sri Jempol Health Clinic, Ministry of Health Malaysia, Bandar Seri Jempol, Negeri Sembilan, Malaysia, 7 Department of General Surgery, Hospital Raja Perempuan Zainab II, Ministry of Health Malaysia, Kota Bahru, Kelantan, Malaysia, 8 Department of General Surgery, Hospital Tuanku Ja'afar, Ministry of Health Malaysia, Seremban, Negeri Sembilan, Malaysia, 9 Department of Medicine, Hospital Serdang, Ministry of Health Malaysia, Kajang, Selangor, Malaysia, 10 Department of Medicine, Hospital Sultanah Aminah, Ministry of Health Malaysia, Johor Bahru, Johor, Malaysia, 11 Department of Medicine, Hospital Raja Perempuan Zainab II, Ministry of Health Malaysia, Kota Bahru, Kelantan, Malaysia, 12 Department of Medicine, Hospital Tengku Ampuan Rahimah, Ministry of Health Malaysia, Klang, Selangor, Malaysia, 13 Family Health Development Division, Ministry of Health Malaysia, Putrajaya, Malaysia, 14 Medical Development Division, Ministry of Health Malaysia, Putrajaya, Malaysia, 15 Institute for Health Systems Research, National Institutes of Health, Director's Office, Ministry of Health Malaysia, Shah Alam, Malaysia

* chong.dwq@moh.gov.my

**Data Availability Statement:** Data used in this study are sourced from the Ministry of Health

## Abstract

### Introduction

Colorectal cancer is a growing global health concern and the number of reported cases has increased over the years. Early detection through screening is critical to improve outcomes for patients with colorectal cancer. In Malaysia, there is an urgent need to optimize the colorectal cancer screening program as uptake is limited by multiple challenges. This study aims to systematically identify and address gaps in screening service delivery to optimize the Malaysian colorectal cancer screening program.

### Methods

This study uses a mixed methods design. It focuses primarily on qualitative data to understand processes and strategies and to identify specific areas that can be improved through

Malaysia and are not publicly available due to confidentiality restrictions. However, data can be accessed upon reasonable request and subject to approval from the relevant authority within the Ministry of Health Malaysia.

**Funding:** This research is funded by the Ministry of Health Malaysia Research Grant [Grant reference number: NIH/800-3/2/2 Jilid 16 (19)]. The funding body is not involved in the study design, data collection and analysis, publication decision, or manuscript preparation.

**Competing interests:** The authors have declared that no competing interests exist.

stakeholder engagement in the screening program. Quantitative data play a dual role in supporting the selection of participants for the qualitative study based on program monitoring data and assessing inequalities in screening and program implementation in healthcare facilities in Malaysia. Meanwhile, literature review identifies existing strategies to improve colorectal cancer screening. Additionally, the knowledge-to-action framework is integrated to ensure that the research findings lead to practical improvements to the colorectal cancer screening program.

## Discussion

Through this complex mix of qualitative and quantitative methods, this study will explore the complex interplay of population- and systems-level factors that influence screening rates. It involves identifying barriers to effective colorectal cancer screening in Malaysia, comparing current strategies with international best practices, and providing evidence-based recommendations to improve the local screening program.

## Introduction

### Global overview and Malaysia's colorectal cancer landscape

Colorectal cancer is a significant global health concern with high morbidity and mortality rates. It accounts for 10% of global cancer incidence and 9.4% of cancer-related deaths [1]. The number of people affected by colorectal cancer is influenced by human development, population growth, and aging [1,2]. This number is estimated to rise to 3.2 million by 2040 [3]. Early detection through effective screening programs is critical for identifying precancerous or early-stage cancer for timely intervention and treatment. The effectiveness of such screening programs in reducing morbidity and mortality associated with colorectal cancer is well-documented [4,5]. In countries with high screening uptake, such as the Netherlands, which has a 64.8% coverage, there has been a significant decrease in colorectal cancer-related morbidity and mortality, underscoring the imperative of widespread screening implementation [6–8].

Colorectal cancer is the second most common cancer in Malaysia, accounting for 13.5% of all new cancer cases diagnosed between 2012 and 2016 [9]. The incidence of colorectal cancer increases with age and is slightly higher in men (14.8/100,000) than women (11.1/100,000). Although colorectal cancer is highly preventable and treatable through early detection, approximately 70% of colorectal cancer patients in Malaysia were diagnosed at stage III or IV [9]. The Ministry of Health Malaysia initiated its colorectal cancer screening program in 2014 to address this challenge. The program uses an immunological fecal occult blood test (iFOBT), followed by colonoscopy, and targets asymptomatic individuals between 50 and 75. By 2020, 598 Ministry of Health clinics are providing this service [10]. However, resource constraints and other barriers have resulted in phased implementation, with less than 1% of Malaysians within the targeted age group undergoing iFOBT screening annually (hereafter referred to as examination coverage), and only 60% of those referred for colonoscopy after a positive iFOBT outcome proceed with the procedure in the public health sector [referred to as further assessment participantion rates; 11].

Despite the recognized importance and demonstrated effectiveness of colorectal cancer screening in improving long-term survival through early diagnosis, the current screening program in Malaysia is inadequate, with participation rates falling short of international

recommendations. The American Cancer Society National Colorectal Cancer Roundtable sets a benchmark by advocating for screening coverage of at least 80% of the eligible population [12,13]. Among countries in the Asia Pacific region with high prevalence and mortality rates, Japan had an examination coverage of 11%, and Korea had a rate of 29.2%. Further assessment participation rates for these countries were 68.5% for Japan and 33% for Korea [8]. The significant discrepancy between current colorectal cancer screening rates in Malaysia and international benchmarks highlights the urgent need to improve the screening program significantly.

## Systemic challenges: Screening implementation and socioeconomic barriers

Challenges to effective screening can be multifaceted and stem from systemic deficiencies and socioeconomic inequalities [14–16]. Colorectal cancer screening programs rely on individual participation and a well-functioning healthcare system that can effectively deliver screening services [17]. Efficient and streamlined workflows are essential for a successful screening program [18]. However, variations may occur at various levels in the implementation of the screening program, which includes multiple components and phases, such as promotion of the iFOBT, screening test administration, and subsequent referral for colonoscopy [19–21]. The screening process in healthcare systems becomes complex due to a lack of integration, causing coordination and continuity of care issues. Communication gaps, unclear responsibilities, and inadequate follow-up mechanisms hinder the seamless flow of the screening process and result in missed opportunities for early detection and timely intervention [8,14,15,22,23]. Additionally, limited access to colonoscopy services, long waiting times, and lack of coordination between primary care and specialist physicians can lead to delays in timely completion of the diagnostic phase [14,15].

It is also crucial to recognize the significant impact of socioeconomic inequalities on screening rates because early screening can influence disease incidence, prevalence, and outcomes [24–27]. Studies have identified several socioeconomic barriers, such as lower educational attainment, income inequality, and limited access to healthcare facilities, contributing to lower screening rates in disadvantaged populations [16,28]. It is important to recognize that these socioeconomic factors may intersect and influence each other (e.g., individuals with lower education levels may also face financial constraints and limited access to healthcare facilities), exacerbating challenges for disadvantaged populations. By implementing comprehensive strategies that consider multiple socioeconomic factors and their interactions, health systems can improve screening rates among underprivileged populations, reduce disparities, and ensure equitable access to screening services. This, in turn, can lead to early detection of colorectal cancer, better outcomes, and lower colorectal cancer morbidity and mortality in socioeconomically disadvantaged populations.

## Optimizing colorectal cancer screening in Malaysia

The effectiveness of colorectal cancer screening is globally recognized, but in Malaysia, implementing and adopting such programs face unique challenges that require a tailored approach. Previous studies have explored individual and community-level barriers to colorectal cancer screening, including socioeconomic inequalities, cultural diversity, and varied perceptions and beliefs about colorectal cancer and its screening [29–31]. However, there is a lack of research that reviews the systemic delivery of colorectal cancer screening, particularly from the perspective of healthcare providers in Malaysia's healthcare system.

The role of healthcare providers in the effectiveness of colorectal cancer screening programs is critical, and there is a need to address the challenges they face in distributing and delivering

screening services across clinics and hospitals. This includes addressing potential inconsistencies in service delivery and the impact of healthcare systems' structure and processes [11,19–21]. A comprehensive analysis is needed to identify the systemic and operational challenges healthcare providers face in implementing a colorectal cancer screening program.

To bridge this gap, our study proposes a comprehensive analysis of the Malaysian colorectal cancer screening program, focusing on the systemic factors experienced by healthcare providers. This approach seeks to identify areas requiring improvement from a service delivery standpoint and propose targeted strategies to optimize the program's reach among healthcare providers. Our study aims to optimize screening uptake rates through improved healthcare delivery by focusing on healthcare providers' challenges and perspectives. To achieve this, our study has specific objectives:

1. To identify strategies to increase colorectal cancer screening uptake and reduce socioeconomic inequalities in iFOBT and colonoscopy screening.

2. To examine the relationship between colorectal cancer screening needs and healthcare needs, the socioeconomic gradient, and ecobiosocial factors to optimize colorectal cancer screening service delivery.

3. To identify critical factors influencing the colorectal cancer screening program and describe effective steps in the pathways of the multicomponent colorectal screening intervention.

## Methods

### Applying systems thinking: A holistic approach to optimize colorectal cancer screening

In this study, systems thinking provides a framework for understanding and improving colorectal cancer screening program [32,33]. The focus shifts from isolated components of the screening process to the broader system [32]. It is recognized that the colorectal cancer screening program is not isolated but is embedded in larger healthcare systems. This perspective recognizes that the success of the screening program depends on the interactions and relationships between various levels of healthcare and stakeholders involved, including individuals, healthcare organizations, communities, and policymakers. Each level influences and is influenced by others, creating a dynamic and interconnected healthcare system [34]. By recognizing the interconnectedness of different levels of healthcare systems and stakeholders, considering the dynamic nature of the system, and identifying leverage points for interventions, systems thinking enables a comprehensive assessment of the screening program as a basis for developing effective strategies that address system-level challenges and improve colorectal cancer screening outcomes [35,36].

### Mixed methods design

A mixed-methods design is used in this study. We describe the components of our study methodology using the Good Reporting of a Mixed Methods Study (GRAMMS) checklist [37; see Annex 1 in S1 File]. The primary data will be qualitative, delving deep into understanding the intricate processes, strategies for improvement, and stakeholder involvement underpinning the screening program. This data will provide valuable insights into the specific areas in need of improvement (Objectives 3).

Simultaneously, the study will utilize quantitative data in a secondary role, which serves two critical purposes. First, the quantitative data obtained from routine monitoring and evaluation

of the screening program will be used sequentially to select participants for the qualitative component. This data will ensure that the qualitative phase is underpinned by empirical evidence from the program's operational data. Next, the study takes on a concurrent approach in which the quantitative data is used to examine inequality in access to screening services and to identify differences in program implementation across healthcare settings, as outlined in Objective 2.

Meanwhile, literature review serves as the fundamental step in identifying strategies from the existing literature to improve colorectal cancer screening (Objective 1). Through this complex mix of qualitative and quantitative methods, we aim to explore the complex interplay of population- and system-level factors that influence screening rates and develop strategies to align the program with international best practices. Fig 1 illustrates the overall study design. It is important to note that the methods used in this study are described in the following order: literature review, quantitative analysis, and qualitative analysis to ensure clarity and consistency with the research objectives. Although the primary role of qualitative data is emphasized, and the secondary importance of quantitative data is acknowledged, the literature review is considered to be of fundamental importance.

## Literature review

**Purpose and rationale.** The review seeks to systematically map and summarize the existing knowledge on colorectal cancer screening strategies, socioeconomic influences, reported opportunities and challenges, and stakeholder involvement. The review will contribute towards prioritizing strategies or interventions that have demonstrated positive outcomes in similar socio-cultural and healthcare settings. The list of strategies will be discussed with participants during a workshop session. The strategies will undergo further deliberation and review by key stakeholders after the workshop (see Qualitative data).

**Search strategy.** The search will include databases such as PubMed, EMBASE, CINAHL, and Google Scholar. The search will be conducted from September to December 2023, and subsequent updates will be performed to identify any new relevant literature. In addition to the electronic database, we will search the reference lists of identified studies to identify additional relevant sources. Google Web will also be searched to identify any pertinent unpublished reports or grey literature that may provide valuable insights for the review.

The review will Ide grey literature on screening programs and strategies to improve colorectal cancer screening uptake in different countries, including reports on policies, programs, development plans, and stakeholders involved. Data sources will include reports on interventions, policy documents, strategic plans, and operational documents. This approach ensures a comprehensive understanding of colorectal cancer screening practices, strategies, and stakeholder involvement.

The search terms Iinclude key concepts such as "colorectal cancer", "screening", "socioeconomic factors", "inequalities", "access", "utilization", "Malaysia", "low- and middle-income countries", "high-income countries", "stakeholders", and "strategies". The search string will be modified and adapted based on the specific databases that will be searched to ensure comprehensive coverage of the relevant literature.

**Study selection.** Documents will be included if they meet these criteria: (a) colorectal cancer screening intervention targeting the general adult population at risk of colorectal cancer (population), (b) interventions using iFOBT and colonoscopy (strategies), (c) set in or focused on Malaysia, low- and middle-income countries or high-income countries (context), and (d) documents and literature types such as empirical studies, commentaries, editorials, and policy reviews, grey literature, and review documents. Studies designed to improve cancer screening

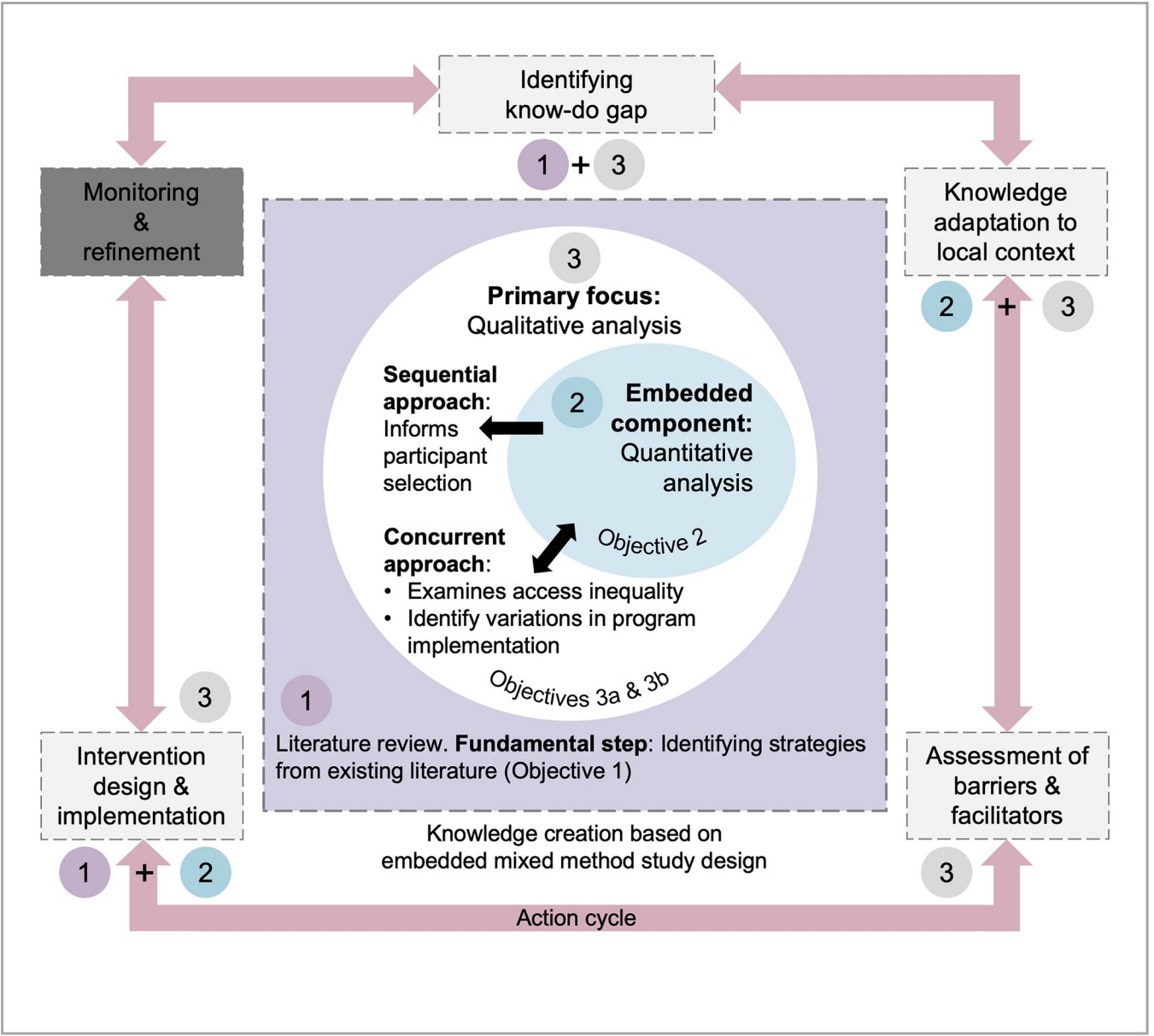

**Fig 1. Overall study design for colorectal cancer screening research.** The central overlapping circles denote the primary qualitative and embedded quantitative data sources. Above this is a rectangle representing the fundamental literature review. The methodological flow of this study is delineated as (1) literature review, (2) quantitative analysis, and (3) qualitative analysis. A larger rectangular surrounding these methods represents the knowledge-to-action cycle. Each segment within this framework represents specific stages of the action process. Key findings from the methods that feed into each phase are highlighted under each phase. Although monitoring and refinement are not detailed in this protocol, they are presented to illustrate the entire cycle from knowledge to action.

in cancer patients and health professionals are excluded, as are intervention protocols if do not include implementation information.

**Data extraction and analysis.** The study employs a two-stage screening approach with an initial assessment of titles and abstracts followed by full-text reviews. It will involve at least two independent reviewers and a pilot study on a sample (10% of documents) will be implemented to refine the protocol. During data charting, information will be extracted using established frameworks such as Template for Intervention Description and Replication checklist with

tools such as Microsoft Excel and ATLAS.ti facilitating the process (provided in Annex 2 in S1 File).

## Quantitative data collection and analysis

**Purpose and rationale.** We aim to understand how colorectal cancer screening rates change across regions and time and what socioeconomic factors might influence these changes. It involves (a) examining the temporal trends in colorectal cancer screening examination coverage and further assessment participation, (b) determining the spatiotemporal associations of colorectal cancer screening with the socioeconomic gradient, healthcare needs, and ecobiosocial factors in Malaysia, and (c) forecasting the screening needs of populations based on predicted cancer screening rates and the effect modification of the socioeconomic gradient.

The cornerstone of this apprIach lies in the fact that inequalities exist within the incidence and screening of cancer, with gradient-like differentials across space and time. These are fundamentally important in predicting healthcare needs within populations. If these estimations or predictions can be made, scenarios can be simulated to inform the allocation of resources within national-level colorectal cancer screening policy.

**Study design and study scope.** A spatiotemporal study will be implemented to examine the associations between the incidence of colorectal cancer screening and area-based indicators of socioeconomic position, which proxy as the upstream determinants of cancer risk factors. This design is chosen because it can capture the dynamic nature of colorectal cancer screening rates over time and allows for various factors' impact across different regions. The study population will include all populations in Malaysia screened for and diagnosed with colorectal cancer in any healthcare facility in Malaysia. The study outcome is colorectal cancer screening rates.

**Data sources and variables.** This study will utilize registry-derived measures of colorectal cancer screening rates in public primary care facilities, census and survey-derived area-based socioeconomic measures, geographic information systems (GIS) data-derived environmental characteristics, and area-based healthcare access characteristics in addressing the study objective. These secondary data will be aggregated from several sources, including routine monitoring and evaluation data of the screening program, national cancer registry, national surveys, and census data from the Department of Statistics Malaysia.

*Individual and community/ population-level characteristics (indicating demand)*. Screening data and colorectal cancer incidence will be sourced from the screening program and self-reported entries in national surveys and the national cancer registry. Socioeconomic indicators will be taken from national surveys and the Department of Statistics Malaysia. Information on physical environments, such as open spaces and land use, will be sourced from the Department of Town and Country Planning.

*Screening services at facility or regional levels (indicating supply)*. Data will include a list of healthcare providers and facilities, especially primary care clinics and their respective referral hospitals. The expansion of colorectal cancer screening services over time will also be examined. A summary of variables that may be included and data sources is provided in Annex 3 in S1 File.

**Data analysis.** Descriptive analysis will be conducted to examine trends and changes in screening rates. Age-standardized rates will be determined and mapped geographically, with visual representations illustrating regional differences. The average annual percentage of change will be estimated to analyze the trends in screening rates over time. It will be analyzed using the piecewise log-linear function of the trend under assumptions of non-linearity within the rate of change [38,39].

Spatial autocorrelation tests, including Moran's I coefficient and Geary's ratio, will measure the clustering effect in the screening rates over the study duration.

Associations between colorectal cancer screening rates and identified factors, such as healthcare resources and ecobiosocial variables, will be analyzed using spatial regression. Specific models within this framework will be selected depending on data characteristics, potentially assuming a Poisson distribution for count or rate data. Additionally, Bayesian spatial regression will be applied using the R-INLA method, with model priors informed by prior knowledge and data nature, detailing hyperparameters and spatial structures, such as the Besag-York-Mollié model. The model specifics, encompassing interaction or higher-order terms, will be refined through preliminary analyses and theoretical underpinnings. Diagnostic checks for spatial and Bayesian spatial regressions will be performed to ascertain model integrity, including residual analysis, spatial autocorrelation checks, and convergence diagnostics like the R-hat statistic for the Bayesian approach.

Variations in screening rates by region will be determined using the most accurate predictive model, and scenario analysis will be employed to observe the potential shifts in screening rates upon altering healthcare resource availability. Sensitivity analyses will be carried out by including and excluding variables to understand the model in greater detail. All data analyses will be conducted using R (version 4.2.2).

## Qualitative data collection and analysis

**Purpose and rationale.**   This study will employ an exploratory qualitative approach to gather stakeholder-specific roles, responsibilities, and critical factors influencing the implementation of colorectal cancer screening intervention in Malaysia. It will involve the development of process flow diagrams and system-support mapping, which will visually depict the process steps and explore how these steps are implemented across different regions, thus capturing procedural aspects and factors influencing the implementation of the colorectal cancer screening program.

**Data collection methods.**   A workshop-style approach will be implemented to explore the participants' perspectives, experiences, and challenges faced during the implementation of the screening intervention. We will examine and map out different steps and processes, identify variations and gaps, and propose improvements in implementing colorectal cancer screening intervention.

Workshops may be followed up with in-depth interviews and observations if additional insights and perspectives from participants regarding the implementation of colorectal cancer screening are required. By actively involving stakeholders, the study aims to ensure that the collected information accurately reflects the realities of implementation and leads to actionable insights for improving screening services.

**Participant selection.**   Participants will be selected using purposive sampling and snowballing techniques. The participants will be recruited from different healthcare facilities (primary care facilities and hospitals), geographic locations (based on regions, i.e., northern, central, southern, east-coast, Sabah, and Sarawak), with varying years of experience, and based on target population's participation rate across screening stages (e.g., examination coverage, hospital appointment adherence, and further assessment participation rates), and colorectal cancer disease burden [9; provided in Annex 4 in S1 File]. The study will involve 50 to 60 participants, including various healthcare professionals and stakeholders involved in implementing the screening program (e.g., medical officers, nurses, medical assistants, laboratory personnel, family medicine specialists, colorectal surgeons, gastroenterologists).

**Group-based activities in workshops.** Direct participant interaction and engagement will be facilitated through two or more in-person workshop sessions. Each workshop will host 25 to 30 participants, with a minimum of five facilitators to steer discussions. These sessions will gather heterogeneous groups of participants to obtain potential variations in the screening process. Additional sessions may be conducted in-person, virtually, or in a hybrid mode, depending on the feasibility and participants' availability, especially if clarification of the process steps is required.

The research team will create a preliminary process map, based on a literature review and input from key stakeholders before the workshops. During the sessions, participants will work in groups of five or six to review this draft. The facilitators will use an interview guide to explore the participants' opinions on the implementation of colorectal cancer screening and identify any areas that could be improved (details in Annex 5 in S1 File). Participants' observations, insights, and suggestions for improvement will be documented on the draft process map during the discussions. The refined process maps from each group will be collected, analyzed, and integrated into the final process flow diagrams to ensure a comprehensive representation of the improved colorectal cancer screening implementation process.

For the second part of the workshop, participants will engage in the concept of system support mapping and receive guidance in creating their system support maps. Participants will be asked to create a support map reflecting their specific colorectal cancer intervention activities. The map should include elements such as (a) their roles within the program, (b) critical responsibilities within each role, (c) needs for effective performance of each responsibility, (d) resources previously utilized for support, and (e) suggestions for individual or systematic support. Additionally, participants will consider (f) contextual factors that may influence their responsibilities and ability to fulfill them, considering differences in populations serviced and geographic regions. Moderators will guide and explain throughout the session to ensure clarity and understanding.

Following the individual mapping exercise, debriefing sessions will be conducted to encourage collective sense-making and reflection on implementation experiences. Participants will have the opportunity to discuss their maps, share insights, and provide feedback, fostering a deeper understanding of the system and individual roles within it. Participants may use papers and sticky notes to create their maps for the systems support mapping activity. The research team will then transfer the maps to mind-mapping software such as Google Jamboard or other whiteboarding platforms, as well as Microsoft Excel or Google Sheets, for further analysis and synthesis.

**Data analysis.** Data collected from the workshops will be analyzed to identify different steps and processes involved in implementing the screening intervention. The analysis will also seek to identify current processes, the variations and gaps, and proposed improvements in the implementation. The process flow diagrams will illustrate the sequential steps in implementing the colorectal cancer screening program and identify key stakeholders associated with each step, such as nurses, medical assistants, medical officers, specialists, laboratory personnel, endoscopy providers, and others.

To analyze the system support maps, we will determine the breadth of entries for each element of the map. We will then use the map entries for each element to identify themes, which will then be compared across the maps. After identifying themes for each element, we will consider the pathways (i.e., connections) between elements, from individual to systems level, with a particular focus on how and why participants selected their proposed improvements. Finally, we will use illustrative quotations to highlight and explain key themes. The analysis will integrate deductive and inductive thematic approaches, as recommended by Braun and Clarke (2006). This approach allows for the structured analysis of predefined themes while remaining open to emerging insights, ensuring a thorough exploration of the data [40].

## Data integration

Data integration will occur at two points: During design and interpretation. Initially, quantitative data derived from routine monitoring and evaluation of the screening program will guide the participant selection for the qualitative component. Once quantitative and qualitative data have been independently gathered and analyzed, we will integrate the findings during interpretation. The trends and patterns observed in colorectal cancer screening uptake over time (quantitative data) will complement and contextualize information gathered from participant discussions (qualitative data).

Given the complexity and multidimensionality of data integration, we recognize that our interpretations may introduce bias. We plan to engage external stakeholders and experts actively to address this issue and ensure a comprehensive and unbiased synthesis of the results. Their valuable input and diverse perspectives will provide an important counterweight to our interpretations, contributing to a more robust analysis.

## Data management and oversight

Access to the data will be limited to the research team members directly involved in data analysis and processing. Strict confidentiality measures will be followed to protect the participants' identities and sensitive information. All data will be anonymized, and unique identifiers or pseudonyms will be assigned to participants to ensure confidentiality during analysis and reporting. The final refined version of the collected data will be stored in a designated repository in the Ministry of Health for seven years after the completion of the research.

## Ethical approval and study registration

The study will comply with ethical principles outlined in the Declaration of Helsinki and Malaysian Good Clinical Practice Guideline. Ethical approval was obtained from the Medical Research and Ethics Committee, Ministry of Health (23-02197-KTT). The study is registered at the National Medical Research Register (NMRR ID-23-02197-KTT). Secondary data used in this study are collected in anonymized format. Data collection will span the study period, expected to be from September 2023 to June 2024 (Objectives 1 and 2). Workshops are scheduled for May/June 2023. Participation in the study is voluntary and participants will provide written informed consent (Objective 3).

## Knowledge translation through the knowledge to action framework

The study findings will be disseminated through established channels of knowledge sharing, such as stakeholder dialogue sessions, peer-reviewed journal articles, and presentations at scientific conferences. Knowledge translation is pivotal in bridging the gap between research and real-world application, ensuring that scientific insights lead to tangible health improvements. In guiding this transition, we apply the Knowledge to Action framework [41], emphasizing the dynamic and iterative relationship between knowledge creation and its practical application (see Fig 1).

**Identifying the know-do gap.** In the initial phase of the knowledge translation cycle, we aim to bridge the gap between existing research evidence and practical applications in the field. Complemented by the qualitative study, our comprehensive literature review lays the foundation for this. By comparing international best practices with current strategies and mapped processes, we will identify discrepancies and areas where improvements are needed in colorectal cancer screening.

**Knowledge adaptation to local context.**   Our quantitative and qualitative analysis provides a detailed view of the current landscape of colorectal cancer screening, providing insights into regional disparities, access challenges, and variations in program implementation. This empirical data serves as a guidepost, allowing us to tailor global best practices to fit the unique nuances and needs of our local context.

**Assessment of barriers and facilitators to knowledge use.**   Our qualitative exploration gathers the perspectives of various stakeholders, identifying systemic, organizational, and individual factors that may either catalyze or impede the implementation of proposed strategies. These findings will be integral in designing context-sensitive interventions and attuning them to on-ground dynamics.

**Intervention design and implementation.**   With a comprehensive understanding from our literature review, quantitative analysis, and qualitative insights, we will collaboratively develop and tailor interventions to improve colorectal cancer screening with key stakeholders. These strategies will address the identified know-do gaps while considering local conditions and potential barriers.

**Monitoring and refinement.**   While a detailed monitoring plan is beyond the scope of this protocol, there will be a concerted effort to monitor how the introduced strategies are being adopted and implemented. Periodic assessment will be conducted to evaluate the uptake of recommendations and the tangible impact on colorectal cancer screening rates. Findings from the study may contribute to areas that need to be monitored and evaluated by program managers in the short term. Future work will involve working closely with stakeholders to refine and revise policies based on feedback and observed outcomes.

This cyclical process will ensure that the improvements in colorectal cancer screening are not transient but become an enduring part of the system. We aim to drive meaningful change that optimizes colorectal cancer screening processes and outcomes by embedding the knowledge translation framework into our research approach.

## Discussion

This article outlines the design and methodological approaches of a multicomponent study aiming to address the issues of suboptimal screening participation in Malaysia's colorectal cancer screening program.

This study's short- and medium-term benefit lies in its potential to streamline the processes and optimize colorectal cancer screening in Malaysia, significantly improving public health outcomes based on a precision public health paradigm. This study will develop strategies that align the program with international best practices suitable to Malaysia's context [11]. It is incumbent upon the study to examine the complex interplay of factors at individual, community, and systemic levels that influence screening rates and explore pragmatic and innovative approaches to overcome these challenges. The identified barriers and challenges within the screening program, including logistical and coordination issues, program complexity, and socioeconomic inequalities, interact with one another. Addressing one aspect may impact others, highlighting the need for a comprehensive and integrated approach to optimize colorectal cancer screening, reduce health inequalities, and reduce the burden of colorectal cancer in the population [42].

Meanwhile, the long-term benefit of this study lies in its potential to position screening as a crucial entry point to the cancer care continuum, extending beyond colorectal cancer. By mapping available resources and understanding the demand for cancer treatment, we can utilize screening as a proxy for access to care. This approach allows us to identify gaps in the healthcare systems and strategically allocate scarce healthcare resources for colorectal cancer

screening based on the population's healthcare needs to ensure early detection and timely and appropriate cancer treatment for those in need.

The significance of this study extends beyond the boundaries of the Malaysian healthcare system. The evidence-based insights and recommendations derived from this study have the potential to offer valuable lessons and pragmatic strategies that can be applied to international settings, particularly in low- and middle-income countries facing similar challenges. Disseminating the study's findings can inform future research endeavors and policy development initiatives, contributing to global advancements in colorectal cancer screening. The significance of this study lies in its contributions to epidemiology through spatiotemporal analysis, its focus on equality and resource allocation, its bottom-up approach through stakeholder engagement, its review of process flow to address challenges, and its emphasis on translating evidence into actions.

The study is not without challenges. First, granularity constraints in spatiotemporal analysis. While spatiotemporal analysis can pinpoint geographic trends and disparities in areas with sparse data or limited reporting, the findings might not capture the complete picture. Additionally, this method captures associations but does not necessarily indicate causation, limiting the depth of insights derived from the quantitative data. This constraint can be addressed by including the qualitative component to understand the barriers, facilitators, and nuances of colorectal cancer screening. Second, while qualitative data provides detailed perspectives and contextual understanding, its inherently subjective nature may limit its applicability to broader populations. However, the insights we will gather and improvements suggested for implementing colorectal cancer screening can be adapted to other settings, provided specific contextual conditions are considered. Third, this study focuses on a colorectal cancer screening program implemented in the public health sector and among healthcare providers. We acknowledge the possibility of concurrent screening initiatives undertaken by the private health sector and nongovernmental organizations. We will include these stakeholders, including end-users in different study phases. This will include sharing key findings through accessible formats like infographics, aiming to capture a wider perspective and mitigate potential oversight.

## Conclusion

In summary, this study combines literature review, quantitative spatiotemporal analysis, and qualitative inquiry to gain a comprehensive understanding of colorectal cancer screening in Malaysia. By contrasting international best practices with local strategies, the study highlights discrepancies and provides data-driven recommendations for strategies tailored to the local context. By applying the knowledge-to-action framework, the study emphasizes continual improvement and aims to optimize colorectal cancer screening implementation and contribute to improving population health outcomes.

## Supporting information

**S1 File. All tables are provided in Annexes 1–5.**
(DOCX)

## Acknowledgments

The authors would like to thank the Director-General of Health Malaysia for permission to publish this paper.

## Author Contributions

**Conceptualization:** Diane Woei-Quan Chong, Vivek Jason Jayaraj, Fathullah Iqbal Ab Rahim, Sharifah Saffinas Syed Soffian, Muhammad Fikri Azmi, Sondi Sararaks.

**Data curation:** Diane Woei-Quan Chong, Vivek Jason Jayaraj, Fathullah Iqbal Ab Rahim, Sharifah Saffinas Syed Soffian, Muhammad Fikri Azmi, Mohd Yusaini Mohd Yusri, Ahmad Shanwani Mohamed Sidek, Norfarizan Azmi, Rosaida Md Said, Muhammad Firdaus Md Salleh, Norasiah Abu Bakar, Hamiza Shahar, Rima Marhayu Abdul Rashid, Shazimah Abdul Samad, Zanita Ahmad, Mohd Safiee Ismail, Adilah A. Bakar, Nor Mashitah Hj Jobli, Sondi Sararaks.

**Formal analysis:** Diane Woei-Quan Chong, Vivek Jason Jayaraj, Fathullah Iqbal Ab Rahim, Sharifah Saffinas Syed Soffian, Muhammad Fikri Azmi, Mohd Yusaini Mohd Yusri, Ahmad Shanwani Mohamed Sidek, Norfarizan Azmi, Rosaida Md Said, Muhammad Firdaus Md Salleh, Norasiah Abu Bakar, Hamiza Shahar, Rima Marhayu Abdul Rashid, Shazimah Abdul Samad, Zanita Ahmad, Mohd Safiee Ismail, Adilah A. Bakar, Nor Mashitah Hj Jobli, Sondi Sararaks.

**Funding acquisition:** Diane Woei-Quan Chong, Vivek Jason Jayaraj, Fathullah Iqbal Ab Rahim, Sharifah Saffinas Syed Soffian, Muhammad Fikri Azmi, Sondi Sararaks.

**Methodology:** Diane Woei-Quan Chong, Vivek Jason Jayaraj, Fathullah Iqbal Ab Rahim, Sharifah Saffinas Syed Soffian, Muhammad Fikri Azmi, Sondi Sararaks.

**Project administration:** Diane Woei-Quan Chong, Vivek Jason Jayaraj, Fathullah Iqbal Ab Rahim, Sharifah Saffinas Syed Soffian, Muhammad Fikri Azmi.

**Supervision:** Sondi Sararaks.

**Writing – original draft:** Diane Woei-Quan Chong.

**Writing – review & editing:** Diane Woei-Quan Chong, Vivek Jason Jayaraj, Fathullah Iqbal Ab Rahim, Sharifah Saffinas Syed Soffian, Muhammad Fikri Azmi, Mohd Yusaini Mohd Yusri, Ahmad Shanwani Mohamed Sidek, Norfarizan Azmi, Rosaida Md Said, Muhammad Firdaus Md Salleh, Norasiah Abu Bakar, Hamiza Shahar, Rima Marhayu Abdul Rashid, Shazimah Abdul Samad, Zanita Ahmad, Mohd Safiee Ismail, Adilah A. Bakar, Nor Mashitah Hj Jobli, Sondi Sararaks.

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
