## [Decision Letter · Decision Letter 0]

7 Jan 2024

PONE-D-23-33266Study protocol for a mixed methods approach to optimize colorectal cancer screening in Malaysia: Integrating stakeholders insights and knowledge-to-action framework.PLOS ONE

Dear Dr. Chong,

Thank you for submitting your manuscript to PLOS ONE. After careful consideration, we feel that it has merit but does not fully meet PLOS ONE’s publication criteria as it currently stands. Therefore, we invite you to submit a revised version of the manuscript that addresses the points raised during the review process.

**Please see the comments of the reviewer for revising the article. **

We look forward to receiving your revised manuscript.

Kind regards,

Abdul Rauf Shakoori

Academic Editor

PLOS ONE

Reviewers' comments:

Reviewer's Responses to Questions

**Comments to the Author**

1. Does the manuscript provide a valid rationale for the proposed study, with clearly identified and justified research questions?

Reviewer #1: Yes

2. Is the protocol technically sound and planned in a manner that will lead to a meaningful outcome and allow testing the stated hypotheses?

Reviewer #1: Partly

3. Is the methodology feasible and described in sufficient detail to allow the work to be replicable?

Reviewer #1: Yes

4. Have the authors described where all data underlying the findings will be made available when the study is complete?

Reviewer #1: Yes

5. Is the manuscript presented in an intelligible fashion and written in standard English?

Reviewer #1: Yes

6. Review Comments to the Author

You may also provide optional suggestions and comments to authors that they might find helpful in planning their study.

Reviewer #1: This protocol for an important study in Malaysia is ambitious and complex. I welcome the opportunity to review the manuscript.

The authors have offered a solid rationale for the study, however they have not offered a view of the literature in relation to other studies that have analysed colorectal screening in Malaysia and have not indicated whether they are addressing a gap in the literature. Additionally the Introduction and background material they have provided is significantly lacking in quality up-to-date references for statements made. References are also required for the concept of 'systems thinking' that has been used in the manuscript.

The authors might consider the use of Rayyan in their literature review.

The authors have reported that the protocol will include a qualitative component to gather insights from key stakeholders in colorectal screening in Malaysia, but have not included consumers as key stakeholders. Additionally the authors need to provide much more detail on the qualitative component of the study in relation to how they conduct meaningful workshops with 50-60 participants, how they will collect data from the participants and how they will analyse these data.

The authors need to indicate how long the study data will be stored.

Though the manuscript is well-written, the authors should review the manuscript for minor spelling and grammar errors.

Finally, there is a reference to timeframes in which the Malaysian government is rolling out its colorectal screening program that seem to be outdated (ie 'by 2020'). Additionally, dates for the scheduling of the study indicate this study is already underway. The authors should review timeframes and dates in the manuscript.

The graphic is useful in clarifying the study for readers.

7. PLOS authors have the option to publish the peer review history of their article (what does this mean?). If published, this will include your full peer review and any attached files.

Reviewer #1: No

---

## [Author Response · Author response to Decision Letter 0]

8 Feb 2024

Dear Editors and Reviewer,

Thank you for your valuable feedback on our manuscript titled “Study protocol for a mixed methods approach to optimize colorectal cancer screening in Malaysia: Integrating stakeholders insights and knowledge-to-action framework.” We have meticulously revised our manuscript in response to your comments, enhancing its clarity, accuracy, and comprehensiveness.

Here are the key revisions made:

1. Literature gap and references. We have updated our introduction with recent references and more explicitly outlined the research gap our study addresses, emphasizing the uniqueness of our approach within the Malaysian context.

2. Systems thinking reference. We have added references to substantiate the use of 'systems thinking' in our study, highlighting its significance in health systems research (refer to point 2 below).

3. Qualitative component detailing. The methodology of our qualitative component has been expanded to detail the execution of workshops for engaging 50-60 participants and our strategies for data collection and analysis.

4. Data storage duration. The duration for data storage has been clarified in accordance with ethical guidelines and research best practices.

5. Minor corrections. All minor spelling and grammar errors identified have been corrected.

6. Timeframes and dates.

– We sought to incorporate the most recent data to reflect the current state of service provision. However, we encountered a limitation in that the latest publicly available and peer-reviewed data on this subject extends only up to the year 2020, as cited in our manuscript [10]. In an effort to address your request, we obtained additional information through personal communication with representatives from the Ministry of Health. These discussions revealed a progressive increment in the number of clinics offering this service beyond 2020, indicating an ongoing expansion effort. Given the nature of this information, we have opted to share these insights with you and the editorial team through this response rather than directly amending the manuscript. This decision is made to maintain the integrity of our references while still acknowledging the dynamic nature of healthcare service provision.

– Regarding the commencement of data collection, we appreciate the opportunity to clarify our position. Publishing our protocol after data collection has begun is a practice recognized for its contribution to research transparency and replicability. This approach allows the scientific community to critically evaluate and replicate our study's methodology, aligning with open science principles. Our protocol provides detailed insights into the innovative methodologies and analytical approaches employed, fostering collaboration and informed discussion. It serves as a foundational reference for future publications, ensuring the evolution of our research design is well-documented.

7. Recommendation the use of protocols.io for study protocol. Thank you for suggesting Protocol.io for our study protocol. We have decided to publish exclusively in PLOS ONE to align with our dissemination strategy and reach our target audience effectively. We appreciate your understanding and support for our decision.

We believe these revisions significantly strengthen our manuscript, making it a valuable contribution to the field. We have attached a point-by-point response to address each comment specifically.

 

1. Line 78. Reference for “It accounts for 10% of global cancer incidence and 9.4% of cancer-related deaths.”

Response: Thank you for your comment. Added a reference to support the statement on global cancer incidence and deaths, now cited in Line 72 “It accounts for 10% of global cancer incidence and 9.4% of cancer-related deaths [1].”

2. Line 79-80. Reference for "The number of people affected by colorectal cancer is influenced by human development, population growth, and aging.”

Response. Thank you. Included references to support the statement on factors influencing colorectal cancer, now in Lines 73-74 “The number of people affected by colorectal cancer is influenced by human development, population growth, and aging [1,2].”

3. Line 88-91. Reference "Colorectal cancer is the second most common cancer in Malaysia, accounting for 13.5% of all new cancer cases diagnosed between 2012 and 2016."

Response: Thank you. Updated with a reference for the prevalence of colorectal cancer in Malaysia, now in Lines 82-83 “Colorectal cancer is the second most common cancer in Malaysia, accounting for 13.5% of all new cancer cases diagnosed between 2012 and 2016 [9].”

4. Line 96-97. Reference to update for this statement “By 2020, 598 clinics are providing this service.”

Response: We sought to incorporate the most recent data to reflect the current state of service provision. However, we encountered a limitation in that the latest publicly available and peer-reviewed data on this subject extends only up to the year 2020, as cited in our manuscript [10]. In an effort to address your request, we obtained additional information through personal communication with representatives from the Ministry of Health. These discussions revealed a progressive increment in the number of clinics offering this service beyond 2020, indicating an ongoing expansion effort. Given the nature of this information, we have opted to share these insights with you and the editorial team through this response rather than directly amending the manuscript. This decision is made to maintain the integrity of our references while still acknowledging the dynamic nature of healthcare service provision.

5. Line 101 – Grammatical error - should be "referred to as…"

Response: Addressed a grammatical error, now corrected in Line 95.

6. Line 106- 108. To update reference because it is dated back 2012.

Response: Strengthened statements with updated references, particularly regarding the American Cancer Society National Colorectal Cancer Roundtable's benchmark, now in Lines 99-101 “The American Cancer Society National Colorectal Cancer Roundtable sets a benchmark by advocating for screening coverage of at least 80% of the eligible population [12,13]”

7. Line 116- Line 120. To add references.

Response: Thank you for the recommendation. Added references to discuss systemic deficiencies and the importance of well-functioning healthcare systems, now in Lines 109-113 “Challenges to effective screening can be multifaceted and stem from systemic deficiencies and socioeconomic inequalities [14–16]. Colorectal cancer screening programs rely on individual participation and a well-functioning healthcare system that can effectively deliver screening services [17]. Efficient and streamlined workflows are essential for a successful screening program [18].”

8. Line 123-124. Grammar check needed "The lack of integration within the healthcare systems adds complexity to the screening process and complicates coordination and continuity of care."

Response: Thank you. We have improved the clarity of the statement:

Line 116-117: “The screening process in healthcare systems becomes complex due to a lack of integration, causing coordination and continuity of care issues.”

9. Line 144. Feedback on subheading “Focus on Malaysia: Beyond the cultural fabric” that it addresses cultural nuances and not going beyond them.

Response: We have improved the subheading "Optimizing colorectal cancer screening in Malaysia" to better reflect the section (Line 137)

10. Line 144: What is needed here is an indication of whether there is any exisiting literature providing an analysis of the malaysian screening program so that readers can understand whether this study fills a gap in the literature or builds on exisiting literature in a more comprehensive way

Response: We have improved the section by providing justification to this study and highlighted that the study focuses on healthcare providers (Line 138-166)

11. Line 165. Provide quality references for systems thinking.

Response: We have provided two references, one of which is the reference that outlines and advocates for using systems thinking to strengthen health systems performance and outcomes. Line 171-172: “In this study, systems thinking provides a framework for understanding and improving colorectal cancer screening program [32,33].”

12. Line 201 - Delete the comma, it's a complicated sentence and the comma here breaks up the flow of the meaning.

Response: We improved the clarity of the statement.

Line 205-209: "Although the primary role of qualitative data is emphasized, and the secondary importance of quantitative data is acknowledged, the literature review is considered to be of fundamental importance.”

13. Line 254: Data extraction and analysis. Consider using Rayyan.

Response: We appreciate the recommendation to use Rayyan for data extraction and analysis. However, due to constraints imposed by our grant funding, we are currently restricted to utilizing the tools that are already available to us.

14. Line 262 - I am not qualified to comment on quantitative methods and tools, however I have reviewed this section of the text for meaning and logic in context to the overall study design.

Response: Thank you, your review of this section for its coherence and alignment with the overall study design is greatly appreciated.

15. Line 361: Participant selection - consumers of screening programs are vital stakeholders in the process.

Response: We acknowledge the importance of including consumers of screening programs as vital stakeholders. In our study, the primary focus is on healthcare providers, with an emphasis on understanding processes and implementation factors. Consumer factors are primarily analyzed from a quantitative perspective to identify gaps and inform resource allocation. Furthermore, our literature review incorporates a synthesis of information from the consumer's perspective. These findings will be triangulated to enrich our study with comprehensive insights into consumer viewpoints.

Additionally, we have outlined our study's focus in the last paragraph of the introduction and reiterated it as a study limitation in Lines 545-551. “We acknowledge the possibility of concurrent screening initiatives undertaken by the private health sector and non-governmental organizations. We will include these stakeholders, including end-users in different study phases. This will include sharing key findings through accessible formats like 

16. Line 368. Group based activities. how many participants will be included in each workshop session?

Response: For the qualitative component of our study, we anticipate involving between 50 to 60 participants in total. To accommodate this, we are planning to conduct at least two workshop sessions.

Line 374-376. "Direct participant interaction and engagement will be facilitated through two or more in-person workshop sessions. Each workshop will host 25 to 30 participants, with a minimum of five facilitators to steer discussions."

17. Line 372-374: Inquiry on data collection methods.

Response: We appreciate your query regarding our data collection approach. We have refined our methodology to ensure comprehensive documentation and analysis. Data will be initially recorded on a draft process map and subsequently compiled and synthesized by our research team for thorough examination.

Line 380-398: "The research team will create a preliminary process map, based on a literature review and input from key stakeholders before the workshops. During the sessions, participants will work in groups of five or six to review this draft. The facilitators will use an interview guide to explore the participants' opinions on the implementation of colorectal cancer screening and identify any areas that could be improved (details in Annex 5 in S1 File). Participants' observations, insights, and suggestions for improvement will be documented on the draft process map during the discussions. The refined process maps from each group will be collected, analyzed, and integrated into the final process flow diagrams to ensure a comprehensive representation of the improved colorectal cancer screening implementation process.

18. Line 402. Inquiry about the analysis methods - Using deductive coding according to a codebook designed on the basis of elements and categories identified in the literature review? Is there an intention to use inductive thematic analysis? Please more detail with quality references

Response: We appreciate your request for clarification on our analytical strategy. 

Line 423-426: “The analysis will integrate deductive and inductive thematic approaches, as recommended by Braun and Clarke (2006). This approach allows for the structured analysis of predefined themes while remaining open to emerging insights, ensuring a thorough exploration of the data [40].”

19. Line 410: line 410 - we will "ten" use the map entries… spelling error

Response: Thank you for highlighting the spelling error. We have edited it to “then”

20. Line 435-436: Duration of data storage

Response: Thank you. We have improved the statement “The final refined version of the collected data will be stored in a designated repository in the Ministry of Health for seven years after the completion of the research.” (Line 447-449)

21. Line 443: Data collection is occurring now.

Response: In response to the reviewer's observation regarding the commencement of data collection and the suggestion to amend the timeline within our manuscript, we appreciate the opportunity to clarify our position on the publication of our protocol at this stage. It is indeed accurate that data collection for our study has begun; however, we believe that the publication of our protocol remains both relevant and valuable for several reasons.

Firstly, the publication of study protocols after the initiation of data collection is recognized as a valuable practice within the scientific community. It provides transparency about the research process, allowing for critical evaluation and replication of the study's methodology. This practice aligns with the principles of open science, enhancing the credibility and reproducibility of research findings. Secondly, our protocol offers detailed insights into the innovative methodologies and analytical approaches employed in our study. Sharing this information at this juncture allows the research community to understand the context and preliminary framework of our ongoing study, fostering collaboration and informed discussion even before the final results are available.

Furthermore, the publication of our protocol serves as a foundational reference for future publications arising from this study, ensuring that readers and researchers can trace the evolution of our research design and its adherence to the initial scientific rigor. Lastly, the initiation of data collection prior to protocol publication does not diminish the protocol's value or relevance. Instead, it provides a unique opportunity to reflect on any adaptive changes made in response to preliminary findings or logistical considerations, contributing to the body of knowledge on adaptive study designs and their implementation in real-world research settings.

Therefore, we respectfully suggest that the timing of our protocol's publication, concurrent with the early stages of data collection, is both justifiable and beneficial to the scientific community. We have reviewed the timeframes and dates within our manuscript to ensure clarity and transparency regarding the study's current status.

22. Line 540 - qualitative spatiotemporal analysis

Response: Line 554: We have included :quantitative spatiotemporal analysis” to the text.

---

## [Editor Report · Decision Letter 1]

13 Feb 2024

Study protocol for a mixed methods approach to optimize colorectal cancer screening in Malaysia: Integrating stakeholders insights and knowledge-to-action framework.

PONE-D-23-33266R1

Dear Dr. Chong,

We’re pleased to inform you that your manuscript has been judged scientifically suitable for publication and will be formally accepted for publication once it meets all outstanding technical requirements.

Kind regards,

Abdul Rauf Shakoori

Academic Editor

PLOS ONE
---

## [Editor Report · Acceptance letter]

1 Apr 2024

PONE-D-23-33266R1 

PLOS ONE

Dear Dr. Chong, 

I'm pleased to inform you that your manuscript has been deemed suitable for publication in PLOS ONE. Congratulations! Your manuscript is now being handed over to our production team.

Kind regards, 

on behalf of

Dr. Abdul Rauf Shakoori 

Academic Editor

PLOS ONE